# Current Evidence on Vaccinations in Pediatric and Adult Patients with Systemic Autoinflammatory Diseases

**DOI:** 10.3390/vaccines11010151

**Published:** 2023-01-10

**Authors:** Maria Grazia Massaro, Mario Caldarelli, Laura Franza, Marcello Candelli, Antonio Gasbarrini, Giovanni Gambassi, Rossella Cianci, Donato Rigante

**Affiliations:** 1Department of Translational Medicine and Surgery, Fondazione Policlinico Universitario A. Gemelli IRCCS, 00168 Rome, Italy; 2Emergency Medicine Unit, Fondazione Policlinico Universitario A. Gemelli IRCCS, 00168 Rome, Italy; 3Department of Translational Medicine and Surgery, Catholic University of the Sacred Heart, 00168 Rome, Italy; 4Department of Life Sciences and Public Health, Fondazione Policlinico Universitario A. Gemelli IRCCS, 00168 Rome, Italy

**Keywords:** autoinflammation, autoinflammatory disease, vaccine, familial Mediterranean fever, cryopyrin-associated periodic syndrome, mevalonate kinase deficiency, periodic fever, aphthosis, pharyngitis adenitis (PFAPA) syndrome, interleukin-1, anakinra, canakinumab

## Abstract

Systemic autoinflammatory diseases (SAIDs) are defined by recurrent febrile attacks associated with protean manifestations involving joints, the gastrointestinal tract, skin, and the central nervous system, combined with elevated inflammatory markers, and are caused by a dysregulation of the innate immune system. From a clinical standpoint, the most known SAIDs are familial Mediterranean fever (FMF); cryopyrin-associated periodic syndrome (CAPS); mevalonate kinase deficiency (MKD); and periodic fever, aphthosis, pharyngitis, and adenitis (PFAPA) syndrome. Current guidelines recommend the regular sequential administration of vaccines for all individuals with SAIDs. However, these patients have a much lower vaccination coverage rates in ‘real-world’ epidemiological studies than the general population. The main purpose of this review was to evaluate the scientific evidence available on both the efficacy and safety of vaccines in patients with SAIDs. From this analysis, neither serious adverse effects nor poorer antibody responses have been observed after vaccination in patients with SAIDs on treatment with biologic agents. More specifically, no new-onset immune-mediated complications have been observed following immunizations. Post-vaccination acute flares were significantly less frequent in FMF patients treated with colchicine alone than in those treated with both colchicine and canakinumab. Conversely, a decreased risk of SARS-CoV-2 infection has been proved for patients with FMF after vaccination with the mRNA-based BNT162b2 vaccine. Canakinumab did not appear to affect the ability to produce antibodies against non-live vaccines in patients with CAPS, especially if administered with a time lag from the vaccination. On the other hand, our analysis has shown that immunization against *Streptococcus pneumoniae*, specifically with the pneumococcal polysaccharide vaccine, was associated with a higher incidence of adverse reactions in CAPS patients. In addition, disease flares might be elicited by vaccinations in children with MKD, though no adverse events have been noted despite concurrent treatment with either anakinra or canakinumab. PFAPA patients seem to be less responsive to measles, mumps, and rubella-vaccine, but have shown higher antibody response than healthy controls following vaccination against hepatitis A. In consideration of the clinical frailty of both children and adults with SAIDs, all vaccinations remain ‘highly’ recommended in this category of patients despite the paucity of data available.

## 1. Introduction

Systemic autoinflammatory diseases (SAIDs) are a group of disorders caused by dysregulation of the innate immune system, which triggers an apparently unprovoked inflammation in the absence of autoantibodies or antigen-specific T cells [1,2]. In particular, they are determined by mutations in genes acting on different branches of innate immunity, but protean autoinflammatory mechanisms have been unraveled for a larger group of disorders having a polygenic/multifactorial origin such as gout, pseudogout, type 2 diabetes, different metabolic disorders, asbestosis, silicosis, and Alzheimer’s disease [3]. Genetic screening has indeed improved the quality of life of patients with SAIDs by providing early diagnosis and allowing more appropriate treatments. Among SAIDs, the best-known include familial Mediterranean fever (FMF) [4], cryopyrin-associated periodic syndrome (CAPS) [5], and mevalonate kinase deficiency (MKD) [6]. Among the non-genetic recurrent fevers with an autoinflammatory mechanism the most common is the syndrome characterized by periodic fever, aphthosis, pharyngitis, and cervical adenitis, named PFAPA syndrome [7], which is largely predominant in the pediatric age.

SAIDs result from an aberrant activation of the innate immune system. The initial step is the recognition of pathogen-associated molecular patterns (PAMPs) or danger-associated molecular patterns (DAMPs) by pattern recognition receptors (PRRs), which are mainly expressed on myeloid cells. After ligand binding, PRRs interact with the adaptor protein myeloid differentiation factor 88 (MyD88), with subsequent activation of the transcription nuclear factor kappa-light-chain-enhancer of activated B cells (NF-κB) and mitogen-activated protein kinase (MAPKs). In turn, NF-κB and MAPKs activate the transcription of specific genes by direct binding to chromatin or by activating downstream factors, such as cyclic adenosine monophosphate (cAMP) response element-binding protein (CREB) or activator protein 1 (AP1) [8]. These transcription factors determine epigenetic changes that increase the expression of interleukin (IL)-1 gene products, such as pro-IL-1β and pro-IL-18. The activation of the IL-1 gene leads to an amplification loop involving inflammasomes, i.e., cytosolic multiprotein oligomers that are assembled after recognition of PAMPS and DAMPS. Once inflammasomes are structurally organized, they activate caspase-1, which promotes the secretion of biologically active IL-1β and IL-18 [9]. There are two types of IL-1 receptors: type I is primarily responsible for transmitting IL-1 inflammatory effects, while type II acts as a suppressor of IL-1 activity by competing for IL-1 binding. IL-1, and specifically IL-1β, binds to type I receptors and activates NF-κB-dependent genes, though the exact mechanisms that govern inflammasome-associated signaling pathways are still to be unraveled fully [10,11].

Clinically, SAIDs are characterized by recurrent episodes of fever variably combined with joint signs, gastrointestinal complaint, skin rashes, neurologic manifestations, and organ-specific inflammation that varies depending on the genetic defect. Acute phase reactants such as C-reactive protein and serum amyloid-A are characteristically elevated during the febrile episodes [12]. Regarding treatment, colchicine is usually the first-choice agent in FMF, while the use of IL-1 inhibitors is reserved for colchicine-resistant or intolerant FMF patients [13]. Instead, treatment is based on IL-1 inhibitors for CAPS and MKD [14,15,16,17,18,19,20,21,22]. For patients with PFAPA syndrome treatment is merely symptomatic and focused on low-dose corticosteroids during the febrile attack, while there is limited experience on the use of IL-1 inhibitors for very few adult patients [23,24,25,26,27].

Despite a general agreement about mandatory vaccinations for clinically vulnerable populations, including patients with SAIDs, suboptimal rates of adherence to vaccination programs in clinical practice are common. Concerns about efficacy, immunogenicity, or safety of vaccinations are among the most important determinants of low vaccination coverage. Nevertheless, current guidelines recommend administration of vaccines in pediatric and adult patients with SAIDs during the quiescent phase of illness, including the seasonal influenza vaccination [28,29].

The main purpose of this paper is to evaluate the available scientific evidence about the efficacy and safety of vaccines in patients with SAIDs. To this aim we have performed a medical literature search in PubMed. Studies written in English published in the last 20 years which included “autoinflammatory diseases” in general, but also specifically dealing with FMF, CAPS, MKD and PFAPA syndrome were selected combining the terms “vaccinations” or “vaccines” as keywords. Additionally, a manual search was carried out through referenced articles to expand the number of studies to evaluate. The eligibility assessment was performed by two authors (M.G.M. and M.C.), who independently reviewed all selected studies and discussed them with all coauthors: a total number of 23 papers judged adequate for the primary aim of this paper was extensively analyzed.

## 2. General Data on Vaccinations in Systemic Autoinflammatory Diseases

Several types of vaccines are currently available: live-attenuated vaccines, inactivated vaccines, viral vector vaccines, toxoid vaccines, messenger RNA (mRNA) vaccines, subunit/recombinant/polysaccharide and conjugate vaccines. Live-attenuated vaccines contain a weakened version of viruses and are based on their ability to replicate sufficiently and induce an effective immunologic response [30]. One of the advantages of live-attenuated vaccines is that they reproduce a biological reaction very similar to natural infections. However, not all individuals can safely receive live-attenuated vaccines, especially immunocompromised subjects [31]. Instead, non-live vaccines are more stable and safer than live-attenuated ones. Their antigenic component may consist of killed whole organisms (e.g., whole-cell pertussis vaccine and inactivated polio vaccine), purified proteins from the pathogen (e.g., acellular pertussis vaccine), recombinant proteins (e.g., hepatitis B virus vaccine), or polysaccharides (e.g., pneumococcal vaccine) [32]. Following the severe acute respiratory syndrome coronavirus 2 (SARS-CoV-2) pandemic, we have witnessed a massive growth in nucleic acid-based vaccines; these vaccines consist of either DNA or RNA encoding the target antigen, which induces both humoral and cellular immune responses. They are easily adaptable against possibly emerging pathogens, but, in contrast, they need specific injection devices or carrier molecules to be delivered directly into cells [32,33]. Each vaccine has a specific administration route: a recent meta-analysis has shown that low-dose intradermal influenza vaccination could represent an alternative to the standard intramuscular dose [34], and a recent cohort study compared intradermal fractional doses of the mRNA SARS-CoV-2 vaccine to intramuscular whole doses, finding a similar immunogenicity [35].

In patients with SAIDs, there is no deficiency in the immune responses but rather a dysregulated innate immunity pattern of activation and turning-off that determines apparently unprovoked systemic inflammation. There are no real shortcomings to the use of any type of vaccine in patients with SAIDs, but caution is recommended in the case of patients treated with immunosuppressive drugs [28,29]. More specifically, inactivated vaccines can be administered to these patients, while live-attenuated vaccines should be avoided and 4 weeks should elapse before the start of immunosuppression [28,29,36,37,38]. Vaccines for measles, mumps, and rubella (MMR) and varicella zoster virus (VZV) may represent an exception to this indication [28,29]. Unlike other live-attenuated vaccines, studies are available about MMR and VZV immunizations in patients with rheumatic diseases receiving immunosuppressive and biological therapy. In five studies with a small number of patients with SAIDs [39,40,41,42,43], VZV and MMR vaccines were proved to be safe, without any report of adverse reactions or vaccine-related infections despite treatment with both immunosuppressive and anti-IL-1 agents.

Some vaccines may need adjuvants to enhance the magnitude and durability of the immune response, though they may elicit a pathological condition named autoimmune/inflammatory syndrome induced by adjuvants (or ASIA), which has shown a larger predominance in females and a higher rate after immunization against hepatitis B (HEPB) and influenza [44]. In particular, the mRNA SARS-CoV-2 vaccines do not include adjuvants of any sort, decreasing the probability of any unwanted reactions.

Regarding the administration of non-live vaccines in patients undergoing anti-IL-1 therapies, Chiato et al. evaluated the efficacy of inactivated adjuvated influenza (Agrippal) and meningococcal group C conjugate (MenC) vaccinations in healthy patients exposed (<2 weeks before vaccinations) to a single 300 mg dose of canakinumab (a fully human monoclonal antibody neutralizing IL-1β) [45]. Overall, 51 healthy adults were enrolled: 25 subjects received canakinumab 300 mg 2 weeks before vaccination and 26 were not exposed to the anti-IL-1 agent. In this study, canakinumab did not affect the development of a protective response against influenza and meningitis. Indeed, the response to both vaccines was similar in both canakinumab-treated and control groups as well as the frequency of adverse events. Headache was the most commonly reported side-effect, but no serious adverse events were reported. The authors concluded that a single dose of 300 mg of canakinumab did not affect the immune response to influenza and MenC vaccinations. Despite canakinumab being associated with sustained IL-1 suppression, long-term monitoring of such subjects also excluded the development of adverse events in the long term, especially infections. Unfortunately, no randomized, controlled trials have been performed for patients with SAIDs undergoing either live-attenuated or non-live vaccines, with or without immunosuppressive and biologic therapies. Indeed, the studies available on SAIDs are small, mostly descriptive, and report variable results depending on the type of vaccine used [38,43,46,47,48,49,50,51,52,53,54,55,56,57,58].

## 3. Familial Mediterranean Fever and Vaccinations

FMF is the most common hereditary autoinflammatory disease worldwide and is caused by gain-of-function mutations in the Mediterranean fever (*MEFV)* gene that encodes pyrin, a crucial regulatory protein in innate immunity. Over the last few years, we have witnessed huge progress related to genetic testing and treatment development for FMF. Clinically, although FMF is considered an episodic disease characterized by brief attacks of fever combined with serositis and/or synovitis, recent systematic studies have defined several FMF-associated chronic inflammatory conditions that can worsen its prognosis [4].

A multicenter, retrospective survey conducted by Jeyaratnam et al. included one patient (aged 2 years) who had colchicine-resistant FMF treated with canakinumab. In this patient, who also presented inflammatory bowel disease, the primary health care provider accidentally administered the MMR vaccine; the next dose of canakinumab was delayed until 2 months after vaccination, but no adverse events directly related to the vaccine occurred. However, 1 week after vaccination, the patient experienced an FMF attack with fever and abdominal pain that required hospitalization; during hospital stay the patient was treated with low-dose prednisone and colchicine with a good response [43].

Jensen et al. conducted a prospective study on the response to influenza vaccination in immunocompromised children with rheumatic diseases, including 241 participants aged 6 months to 19 years, of whom 226 had rheumatic diseases and 15 were controls. The vaccines used were inactivated quadrivalent split-virus seasonal influenza vaccines (Vaxigrip or Fluarix). Four children in the cohort were affected by FMF and one was on treatment with an IL-1 inhibitor (drug type and its dosage were not reported); blood samples were collected before vaccination and up to 3 days after vaccination. All patients and controls showed high antibody titers after vaccination, and no differences were found in the percentage of increase between patients and controls [46].

With regard to vaccine-induced protection against SARS-CoV-2 infection (Table 1), patients with FMF have displayed a safety profile similar to that expected in the general population [59]. In particular, Ugurlu et al. evaluated 48 adult patients with FMF on treatment with both canakinumab and anakinra (a recombinant human IL-1 receptor antagonist), 35 of whom were vaccinated with the inactivated CoronaVac and 13 with the mRNA BNT162b2 [47]. In detail, CoronaVac is an inactivated vaccine developed by the Chinese company Sinovac Biotech, having a good safety and efficacy profile in individuals aged 18 and over [60]. Conversely, BNT162b1 is an mRNA-based vaccine developed by the German biotechnology company BioNTech, composed of nucleoside-modified mRNA that encodes a mutated form of the SARS-CoV-2 spike protein encapsulated in lipid nanoparticles [61]. In the study by Ugurlu et al. patients were vaccinated twice, 1 month apart, and blood samples were collected 14 days after each dose. After the second dose, FMF patients vaccinated with BNT162b2 had higher antibody titers compared to those vaccinated with CoronaVac, but there were no statistically significant differences in antibody titers after the first dose. Therefore, anti-IL-1 therapy did not appear to affect the immune response to vaccines against SARS-CoV-2 [47].

Ozdede et al. analyzed patients with FMF (*n* = 247), 53 of whom were receiving anti-IL-1 therapy (the drug was not specified). Among these patients, 90 were vaccinated with the CoronaVac and 157 with the BNT162b1. The post-vaccine SARS-CoV-2 infection rate was significantly lower in those who had received the mRNA-based BioNTech vaccine (3.2%) than CoronaVac (12.2%, *p* < 0.01). However, occurrence of at least one adverse event was significantly more common among those vaccinated with BioNTech than those vaccinated with CoronaVac (83.4% vs. 53.3%). Most adverse reactions to both vaccines were mild to moderate and transient, and most commonly they were arm pain, fatigue, weakness, headache, or back pain. In a small percentage of cases there were also adverse events that required medical attention (3.3% for CoronaVac and 1.9% for BioNTech), including acute peritonitis, myocardial infarction, and exacerbation of pre-existing heart failure [48].

Another study included 161 FMF patients: 72.7% were vaccinated with BNT162b2, while 27.3% (*n* = 44) were vaccinated with CoronaVac. Overall, 96.3% (*n* = 155) were on active therapy: 93.2% (*n* = 150) were receiving colchicine, 5% (*n* = 8) canakinumab, and 16.8% (*n* = 27) anakinra. Adverse events occurred more frequently in patients vaccinated with BNT162b2 than those who received CoronaVac (54.7% vs. 29.9%). The most commonly reported adverse events were fever, malaise, local pain, and arthralgia in both groups. A single patient developed palmar/plantar pustular psoriasis with arthritis and required hospitalization after one dose of BNT162b2. Disease flares within 1 month after immunization occurred more frequently in patients vaccinated with BNT162b2 (22.2% vs. 19.4%) [49].

Shechtman et al. conducted a retrospective study on the safety of mRNA-based BNT162b2 vaccination in 273 patients with FMF, of whom 95% were vaccinated with two doses. Over 93% of patients were regularly treated with colchicine, while 8% were treated with an IL-1 blocker (canakinumab) in addition to colchicine. After vaccination, 60% of patients reported local reactions (pain at injection site, local swelling, and erythema), while systemic adverse events occurred more than 7 days after vaccination, more frequently after the second dose, and resolved within 2 days. The most common systemic side effects were fatigue, muscle pain, and fever. FMF attacks were reported for 68 patients in the first post-vaccination month, but there were no significant differences in the rate of attacks following the first and second dose of vaccines (19% vs. 19.2%). Moreover, post-vaccination flares were significantly less frequent in patients treated with colchicine alone than those treated with both canakinumab and colchicine [50].

## 4. Cryopyrin-Associated Periodic Syndrome and Vaccinations

CAPS is a spectrum of rare heterogeneous diseases with variable severity caused by Nod-like receptor pyrin domain-containing 3 (*NLRP3*) gene mutations, which give rise to fever and protean complex manifestations involving skin, bones, and the central nervous system and caused by IL-1 oversecretion. Early diagnosis and rapid initiation of treatment usually prevent long-term organ damage in the most severe expression of the disease spectrum [5,15]. Therapeutic approaches aimed at blocking IL-1 have been widely assessed both in clinical trials as well as in real-life, while non-IL-1-targeted therapies have been rather unsuccessful [62,63].

A multicenter, retrospective study conducted by Jeyaratnam et al. included four patients with CAPS who had been vaccinated against yellow fever. In three patients on anakinra (aged 58, 28, and 26 years) medication was suspended 3 days before and resumed 3 days after vaccination, and no adverse events or disease flares occurred. The other (44-year-old) patient, who was receiving canakinumab (150 mg/2 months), was vaccinated against yellow fever 8 weeks after the last dose of canakinumab, and the subsequent dose of canakinumab was administered 3 weeks after vaccination. Even in this patient there were no adverse events or disease flares. Finally, the survey included a 12-year-old CAPS patient receiving anakinra who was vaccinated against MMR after treatment discontinuation in the previous 3 days; therapy was resumed 2 weeks after vaccination, and no adverse events or vaccine-related flares were reported [43].

In addition, a preliminary report by Brogan et al. included 17 CAPS children younger than 4 years who were on treatment with canakinumab every 4–8 weeks. These patients received vaccinations against tetanus/diphtheria/pertussis (DTP), *Haemophilus influenzae* (Hib), *Neisseria meningitidis*, influenza (unspecified type of vaccine), HEPB, and *Streptococcus pneumoniae* (pneumococcal polysaccharide vaccine, PPV; pneumococcal conjugate vaccine, PCV). A rise of antibody titers at 28 and 57 days after vaccinations was observed for all vaccines, and no serious adverse events occurred. The authors concluded that canakinumab did not affect the ability to produce antibodies against childhood non-live vaccines [51].

Another study by Kuemmerle-Deschner et al. included adult and pediatric patients (*n* = 166) with CAPS on therapy with canakinumab. During the study some patients received the following vaccines: influenza (unspecified type of vaccine) in 15 (9%) patients, *Streptococcus pneumoniae* in 5 patients (3%), live-attenuated vaccine against MMR in 1 patient. None of the subjects reported pathogen-related infections [52].

Jaeger et al. considered 68 patients with CAPS who had received a total of 159 vaccine injections. The majority of patients (81%) received influenza vaccine (unspecified type of vaccine), 26% of patients received anti-pneumococcal vaccines (in detail, 14 patients the PPV, 2 the PCV, and 2 a non-specified anti-pneumococcal vaccine), 18% of patients received vaccination against DTP, and 16% of patients received other vaccinations. Adverse reactions were observed in 70% of patients vaccinated against *Streptococcus pneumoniae* compared with 7% among those vaccinated for influenza and 17% among those vaccinated for DTP. Reactions after pneumococcal vaccinations were more severe and lasted significantly longer (up to 3 weeks) than other vaccines. Five patients experienced adverse reactions after PPV, which were fever, local skin reactions and cellulitis at the injection site. There was no temporal association between the last canakinumab administration and occurrence of adverse reactions. Finally, this study reported that pneumococcal vaccines might trigger severe local and systemic reactions in CAPS patients compared with other vaccines. However, the study sample was too small to conclude that PCV might be safer than PPV in patients with CAPS [53].

Walker et al. analyzed 7 CAPS patients (2 were children, aged 7 and 10 years) who were vaccinated against *Streptococcus pneumoniae* with PPV (6/7) and PCV (1/7). All patients reported severe local reactions at injection site a few hours after the administration of the vaccine. Two patients required hospitalization for systemic reactions, including fever. Regarding therapy, six of seven patients were on active canakinumab treatment, four patients received a canakinumab dose on the same day of vaccination, and two patients received it 15 days after vaccination. Since not all patients received the canakinumab dose on the same day as the vaccine, adverse reactions appeared to be specific to the pneumococcal antigens and not to the concurrent anti IL-1 agent. Therefore, this study suggests that pneumococcal vaccination, in particular PPV, could trigger severe local and systemic inflammatory reactions in patients with CAPS [54]. Table 2 shows data related to vaccines against *Streptococcus pneumoniae* in CAPS.

## 5. Mevalonate Kinase Deficiency and Vaccinations

MKD is a rare autosomal recessive disorder caused by biallelic loss-of-function mutations in the *MVK* gene, encoding the enzyme mevalonate kinase, which is defined by severe febrile flares associated with different nonspecific manifestations, mimicking a host of common pediatric conditions and usually leading to significant diagnostic delay. Sometimes flares are elicited by vaccinations, and some of these patients might show a characteristic increase in serum IgD level in both febrile and non-febrile periods [6].

A retrospective survey by Jeyaratnam et al. included four patients with MKD. Specifically, two patients (aged 12 and 9 years) were treated with anakinra and were both vaccinated against VZV, and both had disease flares. In the first case, anakinra (100 mg/day) was discontinued 3–4 days before the vaccine and resumed 1 day after, while in the other case, anakinra was not administered during the 2 days before and the day following vaccination. No adverse events occurred. A third MKD patient (aged 4 years) who was taking canakinumab received MMR and VZV vaccines; canakinumab was interrupted 3 months earlier and resumed 3 months after vaccination. Before vaccination the patient had a low-grade fever every week. After vaccination the patient experienced a mild flare with fever, vomiting, diarrhea, and headache requiring the use of acetaminophen and ibuprofen. No adverse events occurred. Finally, a 12-year-old MKD patient in partial remission with anakinra underwent vaccination against MMR: anakinra was discontinued 3 days before vaccination and restarted 4 weeks later. No exacerbations of MKD and no adverse events occurred [43].

Bodar et al. observed three adult patients with MKD who were given anakinra or etanercept when disease flares occurred: in one patient a febrile exacerbation occurred 24–48 h after vaccinations against hepatitis A (HEPA) and DTP; the two vaccines were administered 1 month apart, and similar episodes of disease flare occurred after both immunizations. Etanercept was administered in one case and anakinra in the other. In both cases there was no change in the protective antibody titers. However, a greater clinical response with faster resolution of symptoms was documented after administration of anakinra compared to etanercept. Finally, for all three patients, the authors reported previous flares of disease after different childhood vaccinations, though both types of vaccine and characteristics of flares were not specified [58].

## 6. Periodic Fever, Aphthosis, Pharyngitis, and Adenitis Syndrome and Vaccinations

PFAPA syndrome is the most frequent non-hereditary autoinflammatory disorder in childhood as its onset is usually before 5 years, though reports regarding the disorder in adulthood are increasing. The syndrome is mainly characterized by recurrent stereotyped clinical manifestations occurring with fever. Although its pathogenesis is not clearly understood, a multifactorial basis or a polygenic pattern of susceptibility are most probably involved [7,23,24].

Kraszewska-Głomba et al. studied 31 children with PFAPA syndrome and 20 age-matched controls to assess the response to vaccination against MMR. The percentage of protective antibody concentrations against MMR was significantly higher in the control group (95.4%) than in the PFAPA one (74.2%). Furthermore, 6 children with PFAPA syndrome, in whom disease had started during the first year of age, had significantly lower antibody levels against MMR than patients with a later onset (*n* = 25). None of the study participants experienced any severe adverse reactions after vaccination [55].

Another study by Maritsi et al. investigated the immunogenicity and side-effects of inactivated vaccination against HEPA in PFAPA syndrome: a group of 28 patients (age range: 2–5 years) with PFAPA syndrome and a control group of 76 healthy children received two doses of the inactivated vaccine against HEPA. Blood samples were obtained for serology at enrolment, 1 month after the first and second doses of vaccine, and 12 months after the last dose. One month after the first vaccination, 92.9% of PFAPA patients and 77.6% of controls achieved a protective antibody titer; rates increased to 100% for PFAPA patients and 96.1% for controls 1 month after the second dose. In both groups, antibody titers remained high 12 months after completion of the study. No statistically significant difference was observed between the groups, although PFAPA patients showed a better immune response at all time points during the study [56].

Rollet-Cohen et al. performed a retrospective study to assess immunizations in children and adolescents (age range: 2–19 years) with SAIDs. Overall, 90 patients (45%) had PFAPA syndrome, 52 (26%) FMF, 3 MKD, 1 CAPS, and a total of 14 patients (tumor-necrosis-factor-receptor-associated periodic syndrome: *n* = 2, FMF: *n* = 6, MKD: *n* = 3, undefined recurrent fever: *n* = 3) had a previous history of immunosuppressive or biologic therapies (methotrexate, adalimumab, infliximab, golimumab, long-term corticosteroids, anakinra, and/or canakinumab). At year 2 of age, 80% of patients were immunized for *Streptococcus pneumoniae* with PCV and for DTP, Hib, and meningitis with MenC; 69% were immunized for HEPB; and 50% were immunized for MMR. At 7 years, the immunization status was complete in 28% of patients; MMR vaccine coverage improved from 50 to 91%, but DTP coverage was particularly weak (61%). At 15 years, 100% of patients were immunized against *Streptococcus pneumoniae* with PCV, 94% against MMR, 81% against DTP, and 67% against HEPB. At the last outpatient visit the immunization status was considered complete in only 44% of patients, proving that the overall vaccination coverage was suboptimal in children with SAIDs [57].

## 7. Discussion

SAIDs are complex rare disorders characterized by unchecked activation of the innate immune system, which represent challenging clinical entities due to the pleiotropy of their phenotypes and high rate of misdiagnosis. Unfortunately, no trials have been performed for patients with SAIDs undergoing different vaccinations, with or without specific treatments. Indeed, the few studies available are small, mostly descriptive, and a limitation of the present review is related to the poor number of supervisory reports related to patients with SAIDs undergoing vaccinations.

With regard to live-attenuated and non-live vaccines, the analyzed studies reported a total population of 214 subjects: 5 with FMF (2.3%), 145 with CAPS (67.7%), 5 with MKD (2.3%), and 59 with PFAPA syndrome (27.5%) [43,46,47,48,49,50,51,52,53,54,55,56,57]. Patients with PFAPA syndrome (59), FMF (5), and MKD (4), for a total of 68 patients, were children. One patient with MKD was an adult. Finally, the 145 CAPS patients were both adults and children (86 adults and 38 children, and for the remaining 21 patients the age was not specified). In our analysis of patients with SAIDs, there were no significant adverse events directly related to the administration of live-attenuated or non-live vaccines. In some cases, disease flares were reported after vaccinations (mostly for MKD), but there were no significant differences in antibody titers compared with patients without SAIDs. Furthermore, concurrent anti-IL-1 therapy in patients with FMF, CAPS, and MKD did not affect the occurrence of adverse events or the entity of antibody titers. Table 3 shows the number of patients with SAIDs analyzed by the type of vaccine administered.

Regarding adverse events, current data on *Streptococcus pneumoniae* vaccines are available for patients with CAPS. It is well-established that *Streptococcus pneumoniae* infections are an important cause of morbidity and mortality worldwide, particularly in young children and elderly subjects: the 23-valent PPV is referred for serotypes accounting for approximately 90% of invasive infections in the adult population, while the 13-valent PCV is referred for serotypes accounting for 92% of invasive infections in children under the age of 5. In the study that analyzed 68 CAPS patients, 18 vaccinated against *Streptococcus pneumoniae* had the highest rate of adverse events (70%) compared with those vaccinated for influenza (7%) and DTP (17%). The most severe reactions occurred in patients vaccinated with PPV compared with those vaccinated with PCV [53]. A similar conclusion was reached by another study in which 7 CAPS patients were vaccinated against *Streptococcus pneumoniae*: six of seven received PPV and one received PCV. Serious adverse reactions occurred in all of them [52]. Another study including 17 CAPS patients undergoing PPV or PCV reported adverse events in 23.5% of them [51]. Therefore, the *Streptococcus pneumoniae* vaccine was an exception in comparison with other vaccines, although available data are scanty and apply only to CAPS patients.

Regarding vaccines against SARS-CoV-2, the only available data are related to patients with FMF (*n* = 729); no studies are referred to other SAIDs. All 729 FMF patients were adults (over 18 years) and 159 (22%) were receiving anti-IL-1 therapies. Most FMF patients (*n* = 560, 76.8%) were vaccinated with the mRNA-based BNT162b2 vaccine, while 169 (23.2%) were vaccinated with the CoronaVac one [47,48,49,50]. These studies showed that FMF patients receiving mRNA BNT162b2 had higher antibody titers and lower infection rates than those receiving CoronaVac [48,49]. In detail, Ozdeide et al. reported that 12.2% of patients with FMF developed post-vaccine SARS-CoV-2 infections after the administration of CoronaVac, in contrast to 3.2% of patients receiving the BioNTech vaccine [48]. Guven et al. found that the rate of SARS-CoV-2 infection was 2.3% for patients who were vaccinated with the mRNA-based BNT162b2 vaccine vs. 7% for those vaccinated with the CoronaVac vaccine [49].

At the same time, adverse events occurred more frequently in patients receiving the mRNA-based BNT162b2 vaccine. Ozdeide et al. reported, considering the number of adverse events globally (from the mildest to the most severe requiring hospitalization), 84.3% of adverse events in patients who received the mRNA BNT162b2 vaccine vs. 53.3% in those receiving the CoronaVac vaccine [48]. The peculiar dysregulation of innate immunity in the pathogenesis of FMF may play a role in the exacerbation of the disease and in occurrence of adverse events after immunizations [64]. Furthermore, younger age and gender may further contribute to overreactivity of the immune system and disease exacerbations, which accounts for 57.1% of serious adverse events in FMF [48]. Other studies have specifically disclosed that mRNA vaccines might determine more adverse events compared to inactive and protein subunit vaccines [65]. Guven et al. showed that the rate of adverse events was 54.7% for mRNA-based BNT162b2 FMF recipients vs. 29.9% for CoronaVac FMF recipients [49]. On the contrary, the rate of disease flares was almost comparable in the two groups [48,49]. In detail, Ozdeide et al. found that the rate of disease flares was 13.4% for FMF patients receiving mRNA BNT162b2 vaccine vs. 24.4% for those receiving CoronaVac [48]; Guven et al. found that the rate of disease flares was 22.2% for FMF patients receiving the mRNA BNT162b2 vaccine vs. 19.4% for those receiving CoronaVac [49].

Finally, in one of the studies reviewed, disease flares were reported more frequently in FMF patients on anti-IL1 therapy who had received the mRNA BNT162b2 vaccine than in those on therapy with colchicine alone: 52% vs. 24%, respectively. Similarly, adverse events were also more frequent in FMF patients receiving anti-IL1 therapy than in those receiving colchicine alone: 82% vs. 74%, respectively, for local adverse events and 73% vs. 59% for systemic adverse events [50].

Given the current interest in SARS-CoV-2 vaccination, the study by Rodriguez et al. deserves a mention, although it is not directly related to SAIDs. The authors documented the development of autoinflammatory/autoimmune phenomena in six patients after SARS-CoV-2 BNT162b2 vaccination: in detail, optic neuritis flare, autoimmune hepatitis, leukocytoclastic vasculitis, and urticarial vasculitis. Two patients, on the other hand, presented a worsening of symptoms before vaccination with no clear diagnosis of either autoinflammatory or autoimmune disease: bilateral symmetrical arthralgias were observed in one patient and febrile erythematous plaques of different sizes on the extremities in the other. The authors hypothesized that immune-mediated/autoinflammatory mechanisms induced by both SARS-CoV-2 infection and vaccination against SARS-CoV-2 might be implicated [66].

Another topic concerning SARS-CoV-2 vaccinations is the onset of vaccine-related myocarditis. Hayashi et al. using an animal model revealed that defective NF-κB1 expression can be involved in the development of autoimmune myocarditis, especially after HBV vaccination, and that spontaneous myocarditis following mRNA-based SARS-CoV-2 vaccines should originate from allergic mechanisms and/or immune complex deposition [67]. Myocarditis after SARS-CoV-2 vaccination has been widely reported in the medical literature [68], but none of the reports currently available were referred to patients affected by SAIDs covered by this review (FMF, CAPS, MKD, or PFAPA syndrome). Additionally, in no cited studies involving FMF patients undergoing SARS-CoV-2 vaccines [47,48,49,50] myocarditis was reported as a vaccination-related side effect. In the general population, the reported incidence of SARS-CoV-2-vaccine-related myocarditis is very low (2.13 per 100,000 people) [50,69], and therefore a much larger cohort of patients with FMF is needed to assess the susceptibility of patients with FMF to develop this adverse event. Moreover, experimental animal models of acute myocarditis demonstrated high expression of IL-1 and dramatic responses in terms of decreased myocardial inflammation following administration of IL-1 inhibitors, also revealing a blurred boundary between autoimmunity and autoinflammation [70].

## 8. Conclusions

For both adult and pediatric patients with SAIDs the currently available guidelines recommend a regular administration of vaccines according to national immunization schedules [28,29]. Although these patients are particularly vulnerable and exposed to an increased risk of infections that can be fully prevented, vaccination coverage is still low [29]. Our analysis documented that there are no serious adverse effects in patients with SAIDs receiving both live-attenuated and non-live vaccines, and also antibody titers are comparable to those elicited in healthy populations. This is in contrast with studies demonstrating a decreased immunogenicity of influenza and pneumococcal vaccines in patients with systemic lupus erythematosus, probably due to a general impairment of humoral and cell-mediated immune responses [71]. In addition, concurrent anti-IL-1 therapy does not appear to be associated with a higher incidence of adverse effects or a reduced antibody response in patients with SAIDs. However, the *Streptococcus pneumoniae* vaccine, particularly the PPV, might represent an exception as it was associated with higher incidence of adverse reactions in patients with CAPS. Each individual patient’s risk of developing *Streptococcus pneumoniae* infections should be carefully assessed before deciding to administer this type of vaccine [55]. With respect to vaccines against SARS-CoV-2, few data are available and only related to adult patients with FMF. Altogether, results confirm a good antibody production and an overall lower risk of post-vaccination SARS-CoV-2 infection following the mRNA-based BNT162b2 vaccine.

In this review, even for patients receiving immunosuppressive or biologic therapies, neither serious adverse events nor poor antibody responses were observed that would justify not administering these vaccines. More studies are warranted to confirm both the safety and efficacy of live-attenuated, non-live, and mRNA vaccines for patients with SAIDs. Indeed, despite the current paucity of data, vaccinations remain ‘highly’ recommended according to the national immunization programs in all patients with SAIDs.

## Figures and Tables

**Table 1 vaccines-11-00151-t001:** Data regarding vaccinations against SARS-CoV-2 in patients with familial Mediterranean fever.

Studies	Patients (*n*)	Anti-IL-1 (Canakinumab) (*n*)	Type of Vaccine (*n*. of Patients)	Adverse Events (%)	Disease Flares (%)	Rate of SARS-CoV-2 Infection (%)
Ugurlu et al. [47]	48	48	CoronaVac (35)	unknown	unknown	unknown
			mRNA BNT162b2 (13)			
Ozdede et al. [48]	247	53	CoronaVac (90)	53.3	24.4	12.2
			mRNA BNT162b2 (157)	84.3	13.4	3.2
Güven et al. [49]	161	35	CoronaVac (44)	29.9	19.4	7
			mRNA BNT162b2 (117)	54.7	22.2	2.3
Shechtman et al. [50]	273	23	CoronaVac (0)	_	_	_
			mRNA BNT162b2 (273)	50.4	19.2	unknown

**Table 2 vaccines-11-00151-t002:** Data regarding vaccinations against *Streptococcus pneumoniae* in patients with cryopyrin-associated periodic syndrome.

Studies	Patients (n)	Anti-IL-1 (n)	Type of Vaccine (n. of Patients)	Adverse Events (%)	Disease Flares (%)
Brogan et al. [51]	17	17	PPV or PCV (not specified)	23.5	none
Kuemmerle-Deschner et al. [52]	5	5	PPV or PCV (not specified)	unknown	unknown
Jaeger et al. [53]	18	18	PPV (14)	70 * Most serious reactions after PPV	unknown
			PCV (2)		
			Not specified (2)		
Walker et al. [54]	7	6	PPV (6)	100 * Most serious reactions in 2 patients after PPV	unknown
			PCV (1)		

PPV: pneumococcal polysaccharide vaccine; PCV: pneumococcal conjugate vaccine.

**Table 3 vaccines-11-00151-t003:** Number of patients with systemic autoinflammatory diseases analyzed by type of vaccine received.

Vaccines	FMF	CAPS	MKD	PFAPAs
Live-attenuated vaccines				
Measles, mumps, rubella	1	3	2	31
Yellow fever	-	4	5	-
Non-live vaccines				
Varicella-zoster virus	-	-	3	-
*Neisseria meningitidis*	-	17	-	-
Influenza (Vaxigrip or Fluarix)	4	87	-	-
Tetanus/diphtheria/pertussis	-	29	1	-
*Haemophilus influenzae*	-	17	-	-
Hepatitis B	-	23	-	-
Hepatitis A	-	5	1	28
Pneumococcal conjugate/polysaccharide	-	47	-	-
Vaccines against SARS-CoV-2 infection				
BNT162b2	560	-	-	-
CoronaVac	169	-	-	-

FMF: familial Mediterranean fever; CAPS: cryopyrin-associated periodic syndrome; MKD: mevalonate kinase deficiency; PFAPAs: periodic fever, aphthosis, pharyngitis, and cervical adenitis syndrome.

## Data Availability

Not applicable.

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
