# Peer review of "Current Evidence on Vaccinations in Pediatric and Adult Patients with Systemic Autoinflammatory Diseases"

_vaccines, 2023, doi:10.3390/vaccines11010151_

Round 1

Reviewer 1 Report

It is an exciting and important review of vaccination on autoimmune inflammatory diseases. However, there are some important points that should be commented on in the manuscript. 1) the rate of adverse effects against mRNA vaccines is high. What do the authors consider is the mechanism involved? 2) The vaccine administration route should also be commented on. 3) The response to attenuated vaccines should also be clarified. Finally, I would suggest adding the limitations of the review since it is clear that the number of studies and subjects involved in the study is limited, and maybe that is the reason why there are not many new developments in the field. 

Author Response

Dear Editor-in-Chief of “Vaccines”,

first of all my coauthors and I would like to thank sincerely the reviewer for this opportunity of cooperation, following the submission of the paper “Current evidence on vaccinations in pediatric and adult patients with systemic autoinflammatory diseases” for its possible publication upon “Vaccines”.

RESPONSES TO REVIEWER 1

It is an exciting and important review of vaccination on autoimmune inflammatory diseases. However, there are some important points that should be commented on in the manuscript. 1) the rate of adverse effects against mRNA vaccines is high. What do the authors consider is the mechanism involved? 2) The vaccine administration route should also be commented on. 3) The response to attenuated vaccines should also be clarified. Finally, I would suggest adding the limitations of the review since it is clear that the number of studies and subjects involved in the study is limited, and maybe that is the reason why there are not many new developments in the field.

We thank the reviewer for her/his positive comments: we have addressed the points raised in the text of the paper.

In particular, the rate of adverse effects against mRNA vaccines was considerable, and at page 10 we have written that the dysregulation of innate immunity may play a role for the occurrence of adverse events after immunizations in this peculiar category of patients. Furthermore, younger age and gender of FMF patients may further contribute to a different reactivity of the immune system, leading to disease exacerbations that are responsible of 57.1% of serious adverse events.

Other studies have also disclosed that mRNA vaccines may determine more adverse events compared to inactive, viral vector, and protein subunit vaccines; in particular, two novel references have been added accordingly.

In relationship to vaccine administration route we have clarified at page 3 that “Each vaccine has a proper administration route: a recent metanalysis has shown that low-dose intradermal influenza vaccination could represent an alternative to the intramuscular standard-dose, while a recent cohort study has compared intradermal fractional doses of the mRNA SARS-CoV-2 vaccine to intramuscular whole dose, finding a similar immunogenicity”. 

With reference to attenuated immunizations, we have written that vaccines for measles, mumps, rubella (MMR) and varicella zoster virus (VZV) represent an exception to the usual caution needed in the case of concurrent immunoupperessant therapy. Indeed, some details are available about MMR and VZV vaccines in patients with rheumatic diseases on immunosuppressive and biological therapies: all studies referred to these vaccines have been discussed in the paper.

Thanks again for all these helping suggestions,

Rossella Cianci and coauthors

Reviewer 2 Report

Authors present a review about safety and efficacy of vaccines in patients with SAIDS, revealing vaccines efficacy and excluding serious events. There are no main areas or flags or weakness and review seems to be complete. References are updated

Author Response

Dear Editor-in-Chief of “Vaccines”,

first of all my coauthors and I would like to thank sincerely the reviewer for this opportunity of cooperation, following the submission of the paper “Current evidence on vaccinations in pediatric and adult patients with systemic autoinflammatory diseases” for its possible publication upon “Vaccines”.

REVIEWER 2

Authors present a review about safety and efficacy of vaccines in patients with SAIDS, revealing vaccines efficacy and excluding serious events. There are no main areas or flags or weakness and review seems to be complete. References are updated.

We thank the reviewer for her/his positive comments and we are grateful for the favourable judgement given to the references used and for her/his recognition of the completeness of our literature search.

Thank you again,

Rossella Cianci and coauthors

Reviewer 3 Report

The manuscript should be revised to include recent findings on the topic, as listed below. Conclusions must include recommendations to researchers and clinicians.

1: Zhong Z, Wu Q, Lai Y, Dai L, Gao Y, Liao W, Feng X, Yang P. Risk for uveitis relapse after COVID-19 vaccination. J Autoimmun. 2022 Dec;133:102925. doi:10.1016/j.jaut.2022.102925. Epub 2022 Oct 4. PMID: 36209692; PMCID: PMC9531657.

2: DiIorio M, Kennedy K, Liew JW, Putman MS, Sirotich E, Sattui SE, Foster G,Harrison C, Larché MJ, Levine M, Moni TT, Thabane L, Bhana S, Costello W,Grainger R, Machado PM, Robinson PC, Sufka P, Wallace ZS, Yazdany J, Gore-MassyM, Howard RA, Kodhek MA, Lalonde N, Tomasella LA, Wallace J, Akpabio A, Alpízar-Rodríguez D, Beesley RP, Berenbaum F, Bulina I, Chock EY, Conway R, Duarte-García A, Duff E, Gheita TA, Graef ER, Hsieh E, El Kibbi L, Liew DF, Lo C, NudelM, Singh AD, Singh JA, Singh N, Ugarte-Gil MF, Hausmann JS, Simard JF, SparksJA. Prolonged COVID-19 symptom duration in people with systemic autoimmune rheumatic diseases: results from the COVID-19 Global Rheumatology Alliance Vaccine Survey. RMD Open. 2022 Sep;8(2):e002587. doi:10.1136/rmdopen-2022-002587. PMID: 36104117; PMCID: PMC9475962.

3: Rodríguez Y, Rojas M, Beltrán S, Polo F, Camacho-Domínguez L, Morales SD,Gershwin ME, Anaya JM. Autoimmune and autoinflammatory conditions after COVID-19 vaccination. New case reports and updated literature review. J Autoimmun. 2022Oct;132:102898. doi: 10.1016/j.jaut.2022.102898. Epub 2022 Aug 24. PMID:36041291; PMCID: PMC9399140.

4: Watad A, Bragazzi NL, McGonagle D, Adawi M, Bridgewood C, Damiani G,Alijotas-Reig J, Esteve-Valverde E, Quaresma M, Amital H, Shoenfeld Autoimmune/inflammatory syndrome induced by adjuvants (ASIA) demonstrates distinct autoimmune and autoinflammatory disease associations according to the adjuvant subtype: Insights from an analysis of 500 cases. Clin Immunol. 2019Jun;203:1-8. doi: 10.1016/j.clim.2019.03.007. Epub 2019 Mar 25. PMID: 30922961.

5: Murdaca G, Orsi A, Spanò F, Puppo F, Durando P, Icardi G, Ansaldi F.Influenza and pneumococcal vaccinations of patients with systemic lupus erythematosus: current views upon safety and immunogenicity. Autoimmun Rev. 2014Feb;13(2):75-84. doi: 10.1016/j.autrev.2013.07.007. Epub 2013 Sep 14. PMID:24044940.

6: Silva CA, Aikawa NE, Bonfa E. Vaccinations in juvenile chronic inflammatorydiseases: an update. Nat Rev Rheumatol. 2013 Sep;9(9):532-43. doi:10.1038/nrrheum.2013.95. Epub 2013 Jul 2. PMID: 23820860.

 Since the literature search was incomplete, the authors missed many key observations. There is nothing new in the focus of this paper, and to add value, it must be an updated review. 

Author Response

Dear Editor-in-Chief of “Vaccines”,

first of all my coauthors and I would like to thank sincerely the reviewer for this opportunity of cooperation, following the submission of the paper “Current evidence on vaccinations in pediatric and adult patients with systemic autoinflammatory diseases” for its possible publication upon “Vaccines”.

REVIEWER 3

The manuscript should be revised to include recent findings on the topic, as listed below.

1: Zhong Z, et al. Risk for uveitis relapse after COVID-19 vaccination. J Autoimmun. 2022;133:102925.

2: DiIorio M, et al. Prolonged COVID-19 symptom duration in people with systemic autoimmune rheumatic diseases: results from the COVID-19 Global Rheumatology Alliance Vaccine Survey. RMD Open. 2022 Sep;8(2):e002587.

3: Rodríguez Y, et al. Autoimmune and autoinflammatory conditions after COVID-19 vaccination. New case reports and updated literature review. J Autoimmun. 2022Oct;132:102898.

4: Watad A, et al. Autoimmune/inflammatory syndrome induced by adjuvants (ASIA) demonstrates distinct autoimmune and autoinflammatory disease associations according to the adjuvant subtype: Insights from an analysis of 500 cases. Clin Immunol. 2019 Jun;203:1-8.

5: Murdaca G, et al. Influenza and pneumococcal vaccinations of patients with systemic lupus erythematosus: current views upon safety and immunogenicity. Autoimmun Rev. 2014Feb;13(2):75-84.

6: Silva CAet al. Vaccinations in juvenile chronic inflammatory diseases: an update. Nat Rev Rheumatol. 2013 Sep;9(9):532-43.

Since the literature search was incomplete, the authors missed many key observations. There is nothing new in the focus of this paper, and to add value, it must be an updated review. Conclusions must include recommendations to researchers and clinicians.

We thank the reviewer for her/his suggestions.

Following the suggestion of the reference by Watad et al we have written at page 3: “Furthermore, some vaccines need adjuvants to enhance the magnitude and durability of the immune response, though they may elicit a pathological condition named autoimmune/inflammatory syndrome induced by adjuvants (or ASIA), which has shown a larger predominance in females and a higher rate after hepatitis B and influenza vaccinations”.

Following the suggestion of the reference by Rodríguez Y et al we have written in the discussion at page 10 that: “Given the current interest for SARS-CoV-2 vaccination, the study by Rodriguez et al. deserves a mention although not directly related to SAIDs. In detail, they documented the development of autoinflammatory/autoimmune phenomena after administration of the SARS-CoV-2 vaccination in 4 patients attending the post-COVID Unit in Bogotà, Colombia. Two further patients, on the other hand, presented a worsening of symptoms before SARS-CoV-2 vaccine with no clear diagnosis of either autoinflammatory or autoimmune disease. We have also written that: “The authors hypothesized that autoinflammatory/immune-mediated mechanisms induced by both SARS-CoV-2 infection and vaccination against SARS-CoV-2 might be implicated”.

Following the suggestion of the reference by Murdaca et al, at pages 10-11, we have highlighted that immune responses observed after vaccinations were good in the totality of patients with autoinflammatory diseases, adding that: “This is contrast with studies demonstrating a decreased immunogenicity of influenza and pneumococcal vaccines in patients with systemic lupus erythematosus. This is probably due to the well-known impairment of humoral and cell-mediated immune responses in patients with lupus.

We have also considered the reference by Silva et al, adding a new further reference by Bodar et al, recounting all patients included in the review. We have enriched some considerations in the text of our paper, as that reference was actually missed in our initial literature search related to patients with autoinflammatory disorders.

Unfortunately, the surveillance study by Zhong Z et al dealing with cases of uveitis encountered after COVID-19 vaccination was not cited, as there was no possible relation with our review and its goal. Similarly, the paper by DiIorio M et al dealing with prolonged COVID-19 symptom duration in people with systemic autoimmune rheumatic diseases was not cited, as patients reported in that paper had rheumatoid arthritis, systemic lupus erythematosus, inflammatory myositis and Sjögren’s syndrome. No patient was affected with autoinflammatory diseases.

We hope sincerely that the reviewer might understand our honest observations, thanking her/him again for all the suggestions to improve the quality of our manuscript.

Rossella Cianci and coauthors

Reviewer 4 Report

About 50 years ago, it was reported that a man in his twenties developed myocarditis after vaccination with a smallpox vaccine created by attenuated the vaccinia virus (1). After that, the onset of myocarditis has been sporadically reported after various vaccinations. Since the onset of myocarditis has been observed after vaccination with attenuated virus or others as the main component, it is possible that the etiology of myocarditis observed after vaccination with mRNA based COVID-19 vaccine is not SARS-CoV-2 mRNA. The described myocarditis cases had usually occurred 10 to I4 days after a primary vaccination with Smallpox vaccine. Furthermore, in many cases, the onset of myocarditis is observed within 5 days after vaccination with the mRNA based COVID-19 vaccine (2). These time course suggests an allergic mechanism, possibly due to formation of immune complexes. In addition, Autoimmune mechanism is proposed for cardiac-related adverse reactions following human papillomavirus (HPV) vaccination (3).

Due to limited data on the efficacy and safety of COVID-19 mRNA vaccination in patients with autoimmune diseases including collagen disease and rheumatic diseases, mRNA based COVID-19 mRNA vaccination for patients with autoimmune diseases or people with allergic predisposition should be carefully considered.

The authors discuss various side effects seen after vaccination with the COVID-19 vaccine in people with underlying autoimmune diseases. However, study using mice with a genetic background for autoimmune diseases have revealed the development of autoimmune diseases such as myocarditis after vaccination with various vaccines (4). In clinical practice, the onset of autoimmune diseases such as myocarditis and anaphylactic shock is observed even after various vaccinations other than the COVID-19 vaccine. Furthermore, the results of various clinical studies have confirmed the effectiveness of anti-IL-1 agents against various side effects observed after vaccination with the COVID-19 vaccine. 

1. Matthews A.W., et al. Br Heart J. 1974 Oct; 36(10):1043-5.

2. Montgomery J, et al. JAMA Cardiol. 2021 Oct 1;6(10):1202-1206

3. Ryabkova VA, et al. Autoimmun Rev. 2019 Apr;18(4):415-425.

4. Hayashi T. et al. Biomedicines. 2022 Jun 18;10(6):1443.

The authors should discuss the differences between other types of vaccines and the COVID-19 vaccine in side effects seen after vaccination.

Author Response

Dear Editor-in-Chief of “Vaccines”,

first of all my coauthors and I would like to thank sincerely the reviewer y for this opportunity of cooperation, following the submission of the paper “Current evidence on vaccinations in pediatric and adult patients with systemic autoinflammatory diseases” for its possible publication upon “Vaccines”.

REVIEWER 4

About 50 years ago, it was reported that a man in his twenties developed myocarditis after vaccination with a smallpox vaccine created by attenuated the vaccinia virus (Matthews AW et al. Br Heart J 1974). After that, the onset of myocarditis has been sporadically reported after various vaccinations. Since the onset of myocarditis has been observed after vaccination with attenuated virus or others as the main component, it is possible that the etiology of myocarditis observed after vaccination with mRNA based COVID-19 vaccine is not SARS-CoV-2 mRNA. The described myocarditis cases had usually occurred 10 to I4 days after a primary vaccination with Smallpox vaccine. Furthermore, in many cases, the onset of myocarditis is observed within 5 days after vaccination with the mRNA based COVID-19 vaccine (Montgomery J et al. JAMA Cardiol 2021). These time course suggests an allergic mechanism, possibly due to formation of immune complexes. In addition, Autoimmune mechanism is proposed for cardiac-related adverse reactions following human papillomavirus (HPV) vaccination (Ryabkova VA et al. Autoimmun Rev 2019). Due to limited data on the efficacy and safety of COVID-19 mRNA vaccination in patients with autoimmune diseases including collagen disease and rheumatic diseases, mRNA based COVID-19 mRNA vaccination for patients with autoimmune diseases or people with allergic predisposition should be carefully considered. The authors discuss various side effects seen after vaccination with the COVID-19 vaccine in people with underlying autoimmune diseases. However, study using mice with a genetic background for autoimmune diseases have revealed the development of autoimmune diseases such as myocarditis after vaccination with various vaccines (Hayashi T et al. Biomedicines 2022). In clinical practice, the onset of autoimmune diseases such as myocarditis and anaphylactic shock is observed even after various vaccinations other than the COVID-19 vaccine. Furthermore, the results of various clinical studies have confirmed the effectiveness of anti-IL-1 agents against various side effects observed after vaccination with the COVID-19 vaccine. The authors should discuss the differences between other types of vaccines and the COVID-19 vaccine in side effects seen after vaccination.

We thank the reviewer for the completeness of her/his explanations.

The main purpose of our review was simply to update information about vaccines administered in patients with autoinflammatory disorders.

Certainly, we have learnt that the onset of myocarditis may be reported after various vaccinations, and that its driving mechanism may be an immunologic reaction against peculiar vaccine components.

We have enclosed the suggested reference by Hayashi et al and added some considerations dealing with differences between mRNA-based vaccines/other types of vaccines, writing at the end of the Discussion these sentences: “Another topic concerning SARS-CoV-2 vaccinations is the onset of vaccine-related myocarditis. Hayashi et al., using an animal model predisposed to autoimmune diseases, revealed that defective NF-κB1 expression can be involved in the development of autoimmune myocarditis, especially after HBV vaccination. On the other hand, spontaneous myocarditis following mRNA-based SARS-CoV-2 vaccines should originate from allergic mechanisms and immune complex deposition. It is interesting to observe that SARS-CoV-2 vaccinations based on mRNA technology might induce novel autoimmune manifestations in previously healthy individuals and that side effects might occur in people with SAIDs. Overall, the mechanism through which mRNA-based vaccines allow antigen production and prime innate immunity is probably the reason why such patients experience more side effects”.

We hope sincerely that the reviewer might consider valid these observations, thanking her/him again for suggesting these intriguing clues aimed at improving the overall quality of our manuscript.

Rossella Cianci and coauthors

Round 2

Reviewer 3 Report

You done an update that makes it a current review article and a summary document.

Author Response

Dear Editor-in-Chief of “Vaccines”,

first of all, my coauthors and I would like to thank sincerely the reviewer for this opportunity of cooperation, following the submission of the paper “Current evidence on vaccinations in pediatric and adult patients with systemic autoinflammatory diseases” for its possible publication upon “Vaccines”.

We thank the reviewer for her/his positive comments and we are grateful for the favourable judgement.

Thank you again,

Rossella Cianci and coauthors

Reviewer 4 Report

Authors should carefully examine the content of reviewers' questions and comments and respond appropriately.

The authors discuss various side effects seen after vaccination with the COVID-19 vaccine in people with underlying autoimmune diseases. However, study using mice with a genetic background for autoimmune diseases have revealed the development of autoimmune diseases such as myocarditis after vaccination with various vaccines. In clinical practice, the onset of autoimmune diseases such as myocarditis and anaphylactic shock is observed even after various vaccinations other than the COVID-19 vaccine. Furthermore, the results of various clinical studies have confirmed the effectiveness of anti-IL-1 agents against various side effects observed after vaccination with the COVID-19 vaccine.

The authors should discuss the differences between other types of vaccines and the COVID-19 vaccine in side effects seen after vaccination.

The authors do not provide useful medical information to the reader through this review. Authors should submit this review to other journals.

Author Response

Dear Editor of Vaccines,

my coauthors and I would like to thank sincerely the reviewers for the opportunity to improve the overall quality of the manuscript entitled “Current evidence on vaccinations in pediatric and adult patients with systemic autoinflammatory diseases” for the possible publication upon “Vaccines”.

More specifically, these are our responses to the fourth reviewer: Authors should carefully examine the content of reviewers' questions and respond appropriately. The authors discuss various side effects seen after vaccination with the COVID-19 vaccine in people with underlying autoimmune diseases. Study using mice with a genetic background for autoimmune diseases have revealed the development of autoimmune diseases such as myocarditis after vaccination with various vaccines. In clinical practice, the onset of autoimmune diseases such as myocarditis and anaphylactic shock is observed even after various vaccinations other than the COVID-19 vaccine. Furthermore, the results of various clinical studies have confirmed the effectiveness of anti-IL-1 agents against various side effects observed after vaccination with the COVID-19 vaccine. The authors should discuss the differences between other types of vaccines and the COVID-19 vaccine in side effects seen after vaccination. Mice with a genetic background for ‘autoimmune diseases’ have revealed the development of autoimmune diseases such as myocarditis after vaccination with various vaccines, which might respond to IL-1 inhibitors.

Responses to the fourth reviewer

We sincerely thank the reviewer hor her/his criticism, though we need to highlight that our review deals with autoinflammatory diseases, related to dysregulation of innate immunitys.

The specific field of our paper was not autoimmunity, in which adaptive immunity and autoantibodies are largely involved. Of course, the link between autoinflammatory diseases and autoimmunity remains unclear, and the fact that IL-1 inhibitors have been used in post-vaccinal manifestations in autoimmune disorders confirms that inflammasome products, such as IL-1, can activate adaptive immunity pathways, justifying their use in the clinical practice.

Unfortunately, there are no studies related to the frank development of myocarditis in patients with autoinflammatory diseases.

In the initial revised version of our paper we inserted a note dedicated to SARS-CoV-2 vaccination-related myocarditis and reported the Hayashi’s study of an animal model predisposed to autoimmune manifestations in which the development of autoimmune myocarditis was observed.

Now, that period has been changed and some new sentences have been added:

A study by Hajjo et al. explores the mechanism by which myocarditis occurs after the administration of different vaccines: cardiac side effects have been observed after smallpox or typhoid fever vaccines, revealing that post-vaccine myocarditis is more frequent for live vaccines and mRNA-based ones.

We also added this new sentence:

Myocarditis after SARS-CoV-2 vaccination has been widely reported, but none of the reports currently available are referred to patients affected by the SAIDs covered by this review (FMF, CAPS, MKD or PFAPA syndrome). However, in no cited studies (47-50) involving FMF patients undergoing SARS-CoV-2 vaccines myocarditis was reported as a vaccination-related side effect. In the general population, the reported incidence of SARS-CoV-2 vaccine myocarditis is very low (2.13 per 100,000 people), therefore a much larger cohort of patients with FMF is needed to assess the susceptibility of patients with FMF to this adverse event.

We have considered also a further reference to highlight the response to IL-1 inhibitors shown by patients having a post-vaccine myocarditis:

Experimental animal models of acute myocarditis demonstrated high expression of IL-1 and dramatic response in terms of decreased myocardial inflammation following administration of IL-1 inhibitors.

These are our honest responses, starting from the consideration that our paper is dealing with autoinflammatory diseases, ultrarare disorders for which there are poor available data.

We hope sincerely that the reviewer might accept our exlanations and its limits, due to the primary purpose of the paper (i.e. to evaluate the scientific evidence about efficacy and safety of vaccines in patients with autoinflammatory diseases).

We want to thank her/him again sincerely for the great attention given to our paper and for the possibility of clarifying such aspects.

With all our best thankufulness,

Rossella Cianci & coauthors

Round 3

Reviewer 4 Report

Accept in present form